# Identification of a Small Molecule Inhibitor of Hyaluronan Synthesis, DDIT, Targeting Breast Cancer Cells

**DOI:** 10.3390/cancers14235800

**Published:** 2022-11-25

**Authors:** Theodoros Karalis, Andrew K. Shiau, Timothy C. Gahman, Spyros S. Skandalis, Carl-Henrik Heldin, Paraskevi Heldin

**Affiliations:** 1Department of Medical Biochemistry and Microbiology, Science for Life Laboratory, Box 572, Uppsala University, SE-751 23 Uppsala, Sweden; 2Small Molecule Discovery Program, Ludwig Institute for Cancer Research, La Jolla, CA 92093, USA; 3Department of Cell and Developmental Biology, School of Biological Sciences, University of California San Diego, La Jolla, CA 92093, USA; 4Laboratory of Biochemistry, Department of Chemistry, University of Patras, 26500 Patras, Greece

**Keywords:** hyaluronan, small molecule inhibitor, hyaluronan synthase, breast cancer, stemness, invasion

## Abstract

**Simple Summary:**

The most aggressive subtype of breast cancer, triple-negative breast cancer, is characterized by an excessive accumulation of hyaluronan in the cancer and its peritumoral stroma, which has been linked to poor prognosis of the patients. Inhibitors of hyaluronan synthesis would thus have a potential clinical value. We have identified the thymidine analog 5′-Deoxy-5′-(1,3-Diphenyl-2-Imidazolidinyl)-Thymidine (DDIT) as a new non-toxic inhibitor of hyaluronan synthesis. DDIT is more potent than the available inhibitor 4-Methylumbelliferone (4-MU), and significantly suppressed the aggressiveness of triple-negative breast cancer cells grown in tissue culture.

**Abstract:**

Breast cancer is a common cancer in women. Breast cancer cells synthesize large amounts of hyaluronan to assist their proliferation, survival, migration and invasion. Accumulation of hyaluronan and overexpression of its receptor CD44 and hyaluronidase TMEM2 in breast tumors correlate with tumor progression and reduced overall survival of patients. Currently, the only known small molecule inhibitor of hyaluronan synthesis is 4-methyl-umbelliferone (4-MU). Due to the importance of hyaluronan for breast cancer progression, our aim was to identify new, potent and chemically distinct inhibitors of its synthesis. Here, we report a new small molecule inhibitor of hyaluronan synthesis, the thymidine analog 5′-Deoxy-5′-(1,3-Diphenyl-2-Imidazolidinyl)-Thymidine (DDIT). This compound is more potent than 4-MU and displays significant anti-tumorigenic properties. Specifically, DDIT inhibits breast cancer cell proliferation, migration, invasion and cancer stem cell self-renewal by suppressing HAS-synthesized hyaluronan. DDIT appears as a promising lead compound for the development of inhibitors of hyaluronan synthesis with potential usefulness in breast cancer treatment.

## 1. Introduction

Breast cancer is a highly heterogenous disease which is classified in several subtypes based on the expression of estrogen receptor-α (ERα), progesterone receptor (PR) and epidermal growth factor receptor 2 (HER2), among other characteristics [1]. Triple-negative breast cancer (ER^−^, PR^−^, HER2^−^) displays high incidence of metastases and poor outcome of the patients [2].

During breast cancer development and progression, extensive extracellular matrix remodeling has been observed [3]. One of the most abundant extracellular matrix components is hyaluronan, a large polysaccharide consisting of alternating di-saccharide units of N-acetyl-glucosamine (GlcNAc) and D-glucuronic acid (D-GlcUA). Hyaluronan is synthesized by three membrane-integrated enzymes, termed hyaluronan synthases (HASes-HAS1, HAS2 and HAS3), that polymerize uridine diphosphate (UDP)-D-GlcNAc and UDP-D-GlcUA to produce polymers of different lengths [4,5,6,7,8]. The HAS isoforms belong to the glycosyltransferase 2-family, which also contains chitin and cellulose synthases; for these enzymes synthesis and membrane translocation of the substrates are coupled events [8,9]. Hyaluronan released into the extracellular space binds several extracellular proteins and interacts with plasma membrane receptors, like CD44 (Cluster of Differentiation 44) and RHAMM (Receptor for Hyaluronan-Mediated Motility), thereby affecting cellular properties, such as migration, invasion and proliferation [10,11].

Hyaluronan synthesis is controlled by transcriptional regulation of HASes [5,12], as well as by post-translational modifications of these enzymes, including O-GlcNAcylation [13], phosphorylation [14] and ubiquitinylation [15,16,17]. The biological functions of hyaluronan are dependent on its molecular size, which is affected by the activities of degrading enzymes, hyaluronidases (HYALs). In human, several hyaluronidases have been identified, including HYAL-1, HYAL-2 [18], TMEM2 (Transmembrane protein 2) [19], HYBID/CEMIP/KIAA1199 (Hyaluronan Binding Protein Involved in Hyaluronan Depolymerization/Cell Migration-Inducing Additionally, Hyaluronan-Binding Protein) [20] and PH-20/SPAM1 (Sperm Adhesion Molecule1) [21].

In breast cancer, hyaluronan accumulation and CD44 overexpression correlate with higher incidence of metastases and poor patient outcomes [22,23,24,25]. Hyaluronan in breast tumors is synthesized both by stromal and cancer cells and creates a hydrated microenvironment that favors proliferation and migration of breast cancer cells [22,26]. Moreover, by binding to its receptors CD44 and RHAMM, hyaluronan activates signaling through ERK1/2 MAP-kinase, AKT and other pathways enhancing breast cancer cell survival and proliferation [27,28]. CD44 can also act as a co-receptor for tyrosine kinase receptors and integrins, regulating their signaling and function, and ultimately promoting migration and invasion [29,30,31]. In addition, hyaluronidases may partially degrade hyaluronan into smaller fragments that have pro-angiogenic effects [32,33] and sustain the inflammatory microenvironment of tumors [22].

The well-established role of hyaluronan in malignancies, and specifically in breast cancer, has prompted the development of inhibitors that target its synthesis. A well characterized small molecule inhibitor of hyaluronan production is 4-methyl-umbelliferone (4-MU). 4-MU is a non-toxic dietary supplement isolated from plants, like chamomile, and is currently being used for the treatment of biliary spasm [34]. 4-MU limits the availability of UDP-GlcUA by capturing GlcUA to form 4-MU-glucuronide, thereby reducing hyaluronan production [34,35,36,37].

Although 4-MU exhibits a number of promising effects on breast cancer cells, its modest potency limits its therapeutic utility [36,37]. Furthermore, a deeper understanding of the biological roles of hyaluronan synthesis is hampered by the lack of potent and specific inhibitors. Therefore, the aim of this study was to identify a new chemical scaffold that could effectively target hyaluronan synthesis. We report the identification of DDIT as a potent inhibitor of hyaluronan synthesis, and provide a characterization of its functional properties.

## 2. Materials and Methods

### 2.1. Cell Culture and Reagents

Hs578T (invasive ductal breast carcinoma), A549 (lung carcinoma), HEK293T and NHLF-2801-1 (normal human lung fibroblasts), were purchased from ATCC. Human breast cancer cell lines, MDA-MB-231-Luciferace^+^-GFP^+^ control and a clone of this cell line that forms bone metastases (MDA-MB-231-BM; bone-seeking clone) were kindly provided by professor P. ten Dijke (University of Leiden, Leiden, The Netherlands).

Human dermal fibroblast cultures (MTS64) were established from biopsies from individuals undergoing breast reduction surgery (Plastic Surgery Department, University Hospital, Uppsala, Sweden), after the approval, as described previously [38], and used between passages 6–10. The cells were cultured in DMEM (Sigma-Aldrich, Cat. No. D5796, St. Louis, MO, USA) containing 10% fetal bovine serum (FBS; Biowest, Biotech-IgG AB, Stockholm, Sweden), supplemented with 100 IU/mL penicillin, 100 μg/mL streptomycin (Gibco, #15140-122, Uppsala, Sweden) and 2 mM L-glutamine (Gibco, #25030-081), at 37 °C in 5% CO_2_. For NHLF-2801-1 cultures, 5 μg/mL insulin (I1882, SIGMA) was also added in the medium.U-251MG (human glioblastoma) [39], were cultured in RPMI-1640 medium (Sigma-Aldrich, Cat. No. R0883, St. Louis, MO, USA) containing 10% FBS, supplemented with 100 IU/mL penicillin, 100 μg/mL streptomycin and 2 mM L-glutamine, at 37 °C in 5% CO_2_.

Medium was changed every 2–3 days and upon reaching approximately 80% confluency, cells were passaged by trypsinization (trypsin-EDTA, 992830, Statens Vetrinärmedicinska Anstalt) for 1 min at room temperature.

4-MU was purchased from SIGMA (#M1508), Hermes-1 (anti-CD44 hyaluronan binding blocking antibody) was a kind gift from professor Sirpa Jalkanen, Universitry of Turku, Finland. Hyaluronan with the size 1000 kDa hyaluronan was a kind gift from Dr Ove Wik (QMed, Uppsala, Sweden) and hyaluronan of 200 kDa from Hylumed Medical (Genzyme, MA, USA). To determine the level of synthesized hyaluronan and staining of cell cultures, we used the hyaluronan binding protein (HABP) domain of aggrecan which binds specifically and essentially irreversible to hyaluronan; the HABP was purified as described [40] and part of this HABP was biotinylated as described previously [38].

### 2.2. RNA Extraction, cDNA Synthesis and RT-qPCR

After different treatments, the cell monolayer was washed twice with ice-cold PBS and the total cellular RNA was extracted using the RNeasy Mini kit (Qiagen, Ref. No. 74106, Hilden, Germany), according to the manufacturer’s instructions. Then, cDNA was prepared by reverse-transcription of one μg of total RNA with iScript cDNA synthesis kit (Biorad, Cat. No. 1708891, Stockholm, Sweden). Real-time qPCR was carried out on a Biorad bcfx96 cycler using KAPA SYBR Fast (Techtum Lab AB, Stockholm, Sweden) in triplicates (95 °C, 2 min; 40 × (95 °C, 10 s; 60 °C, 30 s)). Primer sequences are listed in Appendix A. Target gene expression was normalized to TBP (reference gene) and was calculated as 2^−ΔCT^ (ΔCT = CT (sample mRNA) − CT (TBP mRNA)).

### 2.3. Quantification of Secreted Hyaluronan

The analysis of secreted hyaluronan in conditioned media was performed as described before [38]. Briefly, 100 µL conditioned media from the cell cultures were collected after the appropriate treatments, diluted in 1% BSA in PBS, and transferred to 96-well plates (NUNC Maxisorp, Thermo Scientific, Waltham, MA, USA), pre-coated with 1 µg/mL of HABP in 50 mM carbonate buffer (pH 9.5) overnight at 4 °C. After three washes and blocking in PBS containing 0.5% (*v*/*v*) Tween 20 or 1% BSA (A7030, IgG-free, Sigma Aldrich, Stockholm, Sweden), respectively, the hyaluronan standards (0–100 ng/mL, Q-Med, Uppsala, Sweden) and samples (at appropriate dilutions in blocking solution) were added. After 1 h of incubation at room temperature and washing, 100 µL biotinylated-HABP (b-HABP; 0.5 µg/mL) was added and samples were incubated for 1 h, at room temperature. Following three washes (to remove excess b-HABP), the b-HABP bound to complexes of hyaluronan trapped by the HABP-coated plates were detected by incubation for 1 h with 100 µL streptavidin-HRP (1:1600, Streptavidin-Horse Radish Peroxidase, GE Healthcare, RPN1231, and developed with 100 µL 3, 3′, 5, 5′-tetramethyl-benzidine liquid substrate (TMB, Sigma Aldrich, T4444) per well. The reaction was terminated by addition of 50 µL 2 M H_2_SO_4_ per well and the absorbance at 450 nm was measured with a plate reader. Between each step, the plate was washed three times with 300 µL PBS containing 0.5%Tween per well.

### 2.4. Labelling of Sulfated Glycosaminoglycans

Glycosaminoglycans were detected following the protocol described in (doi:10.1007/978-1-0716-1398-6_12) with minor modifications. Briefly, we seeded 200,000 cells per well in 12-well plates and labelled the sulfated glycosaminoglycans with 100 μCi/mL ^35^S (Na_2_^35^SO_4,_ NEX041H PerkinElmer) in sulfur-free DMEM containing 10% FBS in the presence or absence of 0.04% DMSO or 80 μM DDIT for 24 h. Subsequently, the supernatants were collected, and the cells were lysed in 1% Triton/PBS and centrifuged at 3000 RPM for 5 min, to remove nuclei. Urea buffer (50 mM acetate pH 5.5, 0.2 M NaCl, 0.1% Triton, 6 M urea) was added to the samples which then were incubated with DEAE-Sephacel for 2 h at 4 °C rotating end-over-end. Subsequently the DEAE-Sephacel was washed 5 times with urea buffer and the samples were eluted twice with 2 M NaCl, 0.1% Triton X-100 in PBS. Radioactivity was measured by scintillation counting and normalized to μg of protein.

### 2.5. Staining of Cells with Fluorescein Diacetate and Propidium Iodide

Fluorescein diacetate (FDA) (Sigma-Aldrich, D6883, USA) and propidium iodide (PI) (Sigma-Aldrich, Cat. No. P4170, USA) staining to detect viable and dead cells, respectively, was performed as described in https://ibidi.com/img/cms/support/AN/AN33, accessed on 21 April 2020. Briefly, 50,000 cells per well were seeded in 24-well plates for 24 h in a medium containing 10% FBS followed by overnight starvation. Then, untreated cells or cells treated with DDIT or 4-MU in serum-free media for different time periods were stained with 8 µg FDA/mL or 20 µg PI/mL at room temperature for three minutes in the dark, washed with PBS twice and visualized with fluorescence microscopy. Cell viability was calculated by counting the FDA- (viable green fluorescent cells) and PI-positive (red nuclei PI-stained dead cells) signal among 200–300 cells.

### 2.6. Immunofluorescence Staining

Immunofluorescence staining for hyaluronan and CD44 was performed essentially as described in [41]. Briefly, cells were grown and treated on coverslips, fixed with 3.7% formalin in PBS for 10 min and permeabilized with 0.1% Triton X-100 in PBS for 5 min. After three washes in PBS containing 10% ethanol, quenching was performed by incubation with 200 mM glycine in PBS, followed by blocking with 5% BSA in PBS for 1 h at room temperature. Then, cells were incubated with a monoclonal antibody against CD44 (Hermes-3 that does not affect CD44-hyaluronan interaction; 1 μg/mL) and b-HABP (2 µg/mL) in 1% BSA/PBS overnight at 4 °C. Alexa-Fluor 568-labeled goat anti-mouse antibody [1:1000, #20110, goat anti-mouse (H+L), Biotium CF^TM^ 594] and Alexa-Fluor 488-Streptavidin conjugate (1:1000 diluted in 1% BSA in PBS) were added and the cells were incubated for 1 h at room temperature in the dark. Nuclei were stained with DAPI and the cells were mounted on slides with MOWIOL mounting medium, and visualized by fluorescence microscopy (Nikon 90i). For confocal imaging the samples were visualized by capturing Z-stacks of the samples with ZEISS LSM700 confocal microscope.

### 2.7. Collagen Type I Invasion Assay (Hanging Drop Assay)

The assay was performed as described before [42]. The cells were seeded in 25 µL drops (100,000 cells/drop) in DMEM containing 5% FBS, 20% methyl-cellulose and the appropriate agents, on the cover of a 100 mm Petri dish and incubated for 48 h. Next, 150 µL per well of serum-free media containing collagen type I (final concentration 1.7 mg/mL) was poured into a 48-well plate. The formed spheres were then diluted in the collagen solution and placed on top of the 48-well plates covered with collagen. The plate was incubated for 5 h, at 37 °C in 5% CO_2_, to allow solidification of the collagen solution. Then, fresh media containing 3% FBS with the appropriate agents was added on top of the collagen gel. Images were captured at 0, 24 and 48 h with a phase contrast microscope. Quantification of the images was performed with ImageJ (v.1.53k).

### 2.8. Cell Culture Wound Healing Assay

Hs578T cells (250,000 per well) were seeded in a 12-well plate for 24 h in medium containing 10% FBS, and after 24 h the cells were starved with serum-free media overnight. The next day, the cell monolayer was scratched with a 200 µL pipette tip, washed twice with PBS and serum-free media containing the indicated compounds was added. Images of the wound were captured at 0 and 12 h with a phase-contrast microscope (Axiovert 40 CFL, Zeiss mounted with AxioCam MRC, Carl Zeiss). Quantification of the images was performed with ImageJ [43]. For migration assays of breast cancer cells in a lung microenvironment, normal human lung fibroblasts (NHLF-2801-1) were cultured for 24 h in DMEM with or without 10% FBS in the absence of insulin. The conditioned medium was then added to Hs578T cells subjected to the wound healing assay.

### 2.9. β-Galactosidase Staining

For the evaluation of senescence, β-galactosidase staining was performed. After different treatments, the cells were fixed with 2% formaldehyde and 0.2% glutaraldehyde in PBS for 5 min, at room temperature. Subsequently, the cells were washed twice with PBS and incubated with X-staining solution [1 mg/mL X-Gal (5-Bromo-4-chloro-3-indolyl β-D-galactopyranoside, 9146, SIGMA), 40 mM citric acid/sodium phosphate buffer (pH 6), 5 mM potassium ferricyanide, 150 mM sodium chloride, 2 mM magnesium chloride] overnight, at 37 °C. Then, the monolayers were washed four times with PBS and visualized via microscopy (Axiovert 40 CFL, Zeiss mounted with AxioCam MRC, Carl Zeiss).

### 2.10. Cell Proliferation

Untreated and DDIT-treated cells (2000 cells/well in 12-well plates) were cultured for 3, 6, 9 and 12 days in the serum-free medium; every three days fresh medium without or with DDIT was added. The cell number was then quantified with crystal violet assay as described in [44]. Briefly, the cell monolayer was washed twice with PBS and the cells were stained with 0.5% crystal violet in a solution of 20% methanol in H_2_O for 20 min at room temperature. The excess dye was removed and the plates were air-dried overnight in the dark, at room temperature. Next, the dye was retrieved by destaining with 100% methanol for 20 min. Absorbance was measured at 570 nm with a plate reader (Enspire, ENSPIRE^®^ Multimode reader, PerkinElmer).

### 2.11. Overexpression of HAS Isoforms in HEK293T

Overexpression of HAS1, 2, and 3 were performed as described before [7]. Briefly, 300,000 HEK293T cells were seeded in 6-well plates and cultured in complete media for 24 h. Next, the cells were transfected with 0.25 µg of plasmid DNA (FLAG, HAS1, FLAG-HAS2 or FLAG-HAS3, 6myc or 6myc-HAS2) using lipofectamine 3000 (#L3000001, Lipofectamine™ 3000 Transfection Reagent, Thermo Fischer Scientific), according to manufacturer’s instructions. The cells were cultured for 24 h before proceeding with the different treatments. The protein stability of 6-myc HAS2 was determined as described before [18]. Briefly, cells were lysed in lysis buffer (1% SDS, 50 mM Tris, 150 mM NaCl, 2 mM EDTA, pH 8.0) supplemented with HALT protease and phosphatase inhibitor cocktail (Thermo Fischer Scientific, 1861281), followed by sonication and 10-fold dilution. The lysates were then centrifuged and protein concentration was determined with BCA assay. Aliquots of 1 mg of protein were incubated with 2 μg anti-c-Myc antibody, and the antigen-antibody complexes were precipitated with 50 μL Protein G Dynabeads (#10004D, Thermo-Fischer Scientific). The complexes were sequentially washed twice with RIPA buffer (see Immunoblotting section), once with 500 mM NaCl, and finally once with RIPA buffer, followed by elution of the proteins with 2× SDS buffer and heating (see Immunoblotting section). Samples were resolved with SDS-PAGE and immunoreactive bands detected with immunoblotting utilizing anti-c-Myc antibody. The alignment of human HAS1, 2, and 3 was performed through uniprot.org using Clustal omega program (number of iterations: 5).

### 2.12. Determination of Hyaluronan Size

Hyaluronan size was determined by electrophoresis in agarose gels, as described before in the online protocol (PEGNAC_HA_Size; NHLBI award number PO1HL107147) and in [41].

### 2.13. Immunoblotting

Cells were washed twice with ice-cold PBS and lysed with RIPA buffer (50 mM Tris-HCl, pH 8.0, 150 mM Nacl, 1% NP-40, 0.5% sodium deoxycholate, 0.1% sodium dodecyl sulfate (SDS) and HALT protease and phosphatase inhibitor cocktail (Thermo Fischer Scientific, 1861281). The cell lysates were centrifuged at 13,000 RPM for 10 min, at 4 °C. The pellet was discarded and the protein concentration in the supernatant containing the total protein lysate was determined by BCA assay (Pierce^TM^ BCA protein assay, Reagent A, 23228, Reagent B, 1859078) according to manufacturer’s instructions. The total cell lysate was diluted in 6× SDS sample buffer (0.5 M Tris-HCl, pH 6.8, 30% *w*/*v* glycerol, 10% *w*/*v* SDS, 0.6 M dithiothreitol and 0.012% *w*/*v* bromophenol blue). The lysates containing SDS sample buffer were then heated at 95 °C for 5 min.

Protein lysates (30–50 µg) were separated by SDS-polyacrylamide gel electrophoresis (PAGE) and transferred to nitrocellulose membranes (Amersham^TM^ Protran^TM^ 0.45 µM, GE Healthcare Life Sciences, #10600002) at 100 V for 2 h, at 4 °C. The membranes were then blocked with 5% BSA or 5% milk in Tris buffered saline (TBS) containing 0.1% Tween for 1 h at room temperature, incubated with primary antibodies overnight at 4 °C and subsequently secondary antibodies for 1 h at room temperature. The membranes were then incubated with ECL (Immobilon^®^ Western Chemiluminescent HRP substrate, Millipore, WBKLS0500) for 1 min and the bands were visualized with a CCD camera (ChemiDOC^TM^ MP Imaging system, BIORAD). Between each step the membranes were washed three times with TBS, 0.1% Tween for 5 min. The antibodies utilized are listed in Appendix A.

### 2.14. Cell Cycle Analysis

Cell cycle analysis was performed via analysis of DRAQ7-stained cells (DRAQ7™, Biostatus, #DR70250) with flow cytometry, according to manufacturer’s instructions. Briefly, the cells were harvested by trypsinization, centrifuged and fixed with 80% ethanol on ice for 30 min. The fixative was removed by centrifugation and the pellet was washed once with PBS. Cells were then stained with 10 nM DRAQ7 in PBS for 15 min at room temperature, passed through cell strainers (CellTrics^®^ 50 μm, Sysmex) and analyzed with flow cytometry (BD Accuri C6). Analysis was subsequently performed with FlowJo_v.10.7.1.

### 2.15. HAS2 Knock-Down with siRNA Transfection

For knock-down of *HAS2*, cells were harvested with trypsinization, counted and seeded together with siRNA against HAS2 (Origene, #878836) or scrambled siRNA (DharmaconTM ON-TARGETplusTM Control pool, #D-001810-10-20) as control, according to manufacturer’s instructions. Briefly, siRNAs were diluted in siRNA buffer (Dharmacon, #B002000UB100) or OptiMEM (Gibco, #11058-021), while silenFect (siLenFectTM Lipid Reagent, BIORAD, #170-3361) was diluted in OptiMEM in a different tube. After incubation for 5 min at room temperature, the solutions were mixed, incubated for 20 min at room temperature and subsequently mixed with the cells. Hs578T cells (800,000 per well) were seeded in 6-well plates and grown for 48 h (30 nM final concentration of siRNA). Then, fresh medium containing 10 nM of siRNA was added and the cells were incubated for an additional 24 h. Subsequently, the medium was replaced with fresh complete medium and the cells were grown for an additional 24 h, before proceeding with the experiment.

### 2.16. Mammosphere Culturing

The experiment was performed as described in [45]. Briefly, cells were trypsinized and collected in a 15 mL Falcon tube. The cells were centrifuged at 1000 rpm for 3 min and the cell pellet was resuspended in DMEM/F12 (1:1; GIBCO, 21331-020) medium. The cell aggregates were dispensed by passing the cells through a 25G needle three times. The cells were then counted with trypan blue and 200,000 cells per well were seeded in a low-attachment 6-well plate (Corning, 3471, Costar^®^ 6-well plate, ultra-low attachment surface) in DMEM/F12 containing B27 (B27 supplement 50X, GIBCO, 17504-044), 25 ng/mL epidermal growth factor (EGF) and 25 ng/mL basic fibroblast growth factor (bFGF). Mammospheres were allowed to grow for 6 days without disturbing the plates. Then, the mammospheres were collected by centrifugation, washed with PBS and utilized for immunoblotting or PCR analysis, as described below.

For quantification of mammosphere forming eficiency, 1900 cells per well were seeded in a low-attachment 6-well plate, as above, and incubated for 6 days without disturbing the plates. Then, the spheres with diameter >40 µm were counted. The spheres were then collected and dissociated with mild trypsin (0.05% trypsin-EDTA, #25300-054) and passaging through a 25 G needle three times. Trypsin was removed, the cells were counted and 1900 cells per well were seeded once more. The secondary spheres were allowed to form for 6 days and the spheres were subsequently counted and photographed. Sphere forming efficiency was calculated as the number of spheres per well divided by the number of cells seeded per well and expressed in per cent.

### 2.17. Statistical Analysis

Each experiment was performed at least three times and for the statistical analysis two-tailed Student’s t-test was used to calculate significance. All error bars in the graphics indicate the standard error of the mean (SEM). Statistical significance is depicted as asterisks (* *p* < 0.05; ** *p* < 0.01; *** *p* < 0.001).

## 3. Results

### 3.1. Identification and Characterization of DDIT as a Novel Hyaluronan Synthesis Inhibitor

We undertook a focused screening approach to identify new hyaluronan synthesis inhibitors. Appendix A depicts the chemical structures of 11 synthetic or natural product analogs of thymidine and uridine with the potential to act as competitors for substrate binding (Compounds **1**–**11**). Furthermore, based on the homology of cellulose synthases to HASes, we included two known cellulose synthesis inhibitors in our screening set (Compounds **12**,**13**) (Appendix A). Treatment of Hs578T breast cancer cells with these compounds led to both activation and inhibition of hyaluronan synthesis with the most potent inhibitor being the thymidine analog 5′-Deoxy-5′-(1,3-Diphenyl-2-Imidazolidinyl)-Thymidine (DDIT; Compound **4**) (Figure 1A,B and Appendix A).

In order to further assess the effects of DDIT on hyaluronan synthesis, we selected several cell lines that synthesize hyaluronan and also determined their mRNA expression levels of molecules involved in hyaluronan turnover and signaling, including hyaluronan synthases, hyaluronidases and the hyaluronan receptors CD44 and RHAMM, using RT-qPCR. The breast cancer cell lines Hs578T and MDA-MB-231 (parental line and a bone-seeking clone) expressed high levels of *HAS2* and lower mRNA levels of *HAS1* and *HAS3*. In agreement with our previous studies, the breast cancer cells that metastasize to bone marrow expressed higher level of *HAS2* than the parental cells (Figure 2A,B; [46]). In the lung adenocarcinoma cell line A549, *HAS2* was expressed at high level, but *HAS3* was also expressed to a significant degree (Figure 2C), while the glioblastoma cell line U-251MG expressed *HAS2*, and even higher level of *HAS3* (Figure 2D). Among hyaluronidases, *TMEM2* was expressed at the highest levels in all cancer cell lines, while *CD44s* was the most prominent *CD44* variant expressed (Figure 2). In MTS64 dermal fibroblasts and NHLF-2801-1 lung fibroblasts, *HAS2* was the main hyaluronan synthase gene expressed, and *CD44s* the main hyaluronan receptor (Figure 2E,F). The cancer cell lines mainly expressed *TMEM2* hyaluronidase, whereas the non-transformed cells predominantly expressed its paralog *KIAA1199/CEMIP*, while *TMEM2* was expressed to a lesser extent (Figure 2E,F).

Because of the low solubility of some compounds, the initial screening, which was performed at a concentration of 20 µM, required a final DMSO concentration of 0.2% (Figure 1). DMSO at this concentration was found to reduce hyaluronan synthesis by approximately 50% (data not shown). To further investigate the inhibitory effect of DDIT on hyaluronan production, we prepared a concentrated stock that, at the highest concentration used, required a final DMSO concentration in the medium of only 0.04%. Upon readjusting the DMSO concentration, treatment of Hs578T, A549, and MTS64 cells with various DDIT concentrations yielded only a minor effect at 20 and 40 μM, but a significant suppressive effect on hyaluronan production at 80 μM by about 50% (Figure 3A,C and Appendix A). Furthermore, 80 μM of DDIT exerted a significant inhibition of about 30% on hyaluronan production by breast cancer cells in MDA-MB-231 parental and bone-seeking cultures (Figure 3B). In U-251MG glioma cells concentrations higher than 100 µM of DDIT were needed in order to achieve significant hyaluronan synthesis inhibition (Figure 3D and Appendix A). Importantly, DDIT at 80 µM reduced hyaluronan synthesis to the same extent as the established hyaluronan synthesis inhibitor 4-MU at 1 mM (Figure 3). A dose–response analysis revealed that more than a 10-fold lower concentration of DDIT was needed to achieve a 50% inhibition of hyaluronan synthesis compared to 4-MU (Figure 3E). At the concentrations used, DDIT was found not to be toxic for the tested cell lines (Appendix A).

The suppressive effect of DDIT on hyaluronan synthesis was clearly detected at 12 and 24 h after its addition to Hs578T breast cancer cultures; at earlier time points the secreted amount hyaluronan was low and difficult to detect (Figure 4A) [15]. Since the activity of drugs can be affected by serum [47], we investigated the ability of DDIT to inhibit hyaluronan production in Hs578T cells cultured in various FBS concentrations. The levels of hyaluronan secretion were reduced to the same extent (about 50%) regardless of serum concentration (Figure 4B) indicating that DDIT is not significantly protein bound.

To determine whether DDIT is selective against any HAS isoform, we attempted to overexpress HAS1, 2 and 3 in HEK293T cells. Overexpression of HAS1 did not result in significant induction of hyaluronan in agreement with previous studies [6]. On the other hand, overexpression of HAS2 or HAS3 led to significant upregulation of secreted hyaluronan which was reduced by DDIT (Figure 4C), suggesting that DDIT is active against both HAS2 and HAS3, as expected by their higher similarity (about 70%) through protein sequence alignment (Figure 4D). Of note, the protein stability of HAS2 did not change after treatment with DDIT (Appendix A), and neither was there any change in the size of synthesized hyaluronan (Appendix A). Of note, DDIT was also able to reduce synthesis of sulfated glycosaminoglycans in hs578T cells (Appendix A).

### 3.2. Exposure to DDIT Alters the Organization of Hyaluronan-Enriched Matrices around Hs578T Cells

As shown in Figure 5, immunofluorescence analysis of untreated Hs578T cells revealed cell-associated hyaluronan, and hyaluronan-rich structures weaving over the cells after 24 h of culture. The cable-like hyaluronan structures formed to a much lesser extent after treatment with DDIT, and also after 4-MU treatment, suggesting decreased hyaluronan translocation and secretion. In DDIT- and 4-MU-treated cultures, hyaluronan expression was predominantly seen intracellularly and at plasma membranes sites. Confocal microscopy analysis verified the formation of hyaluronan-rich fibrous structures also after 6 days of culture in the absence of DDIT (Figure 5B and Appendix A). Thus, DDIT and 4-MU may prevent both synthesis and release of newly synthesized hyaluronan.

### 3.3. Effect of DDIT on the Expression of Hyaluronan Synthase, Hyaluronidase and Cell-Surface Receptor Genes

Since DDIT potently suppressed the amount of secreted hyaluronan, we investigated its effect on the mRNA levels of hyaluronan synthases, hyaluronidase and hyaluronan receptors. Treatment of breast cancer cells Hs578T with DDIT induced the mRNA expression of hyaluronan synthases *HAS1* and *HAS3* more than 2-fold, and the *HAS2* transcript to a much lesser degree, while 4-MU had no effect (Figure 6A). On the other hand, DDIT significantly reduced the expression of hyaluronidases *HYAL-2*, *TMEM2* (encoding the main hyaluronidase expressed in this breast cancer cell line) and *KIAA1199/CEMIP*, while it enhanced *HYAL-1* expression. Of note, 4-MU had no effect on the expression of hyaluronidase transcripts (Figure 6B). Treatment with either DDIT or 4-MU resulted in decreased expression of the *CD44s* isoform and *RHAMM* (Figure 6C). Thus, DDIT does not suppress hyaluronan synthesis by inhibiting the expression of HAS mRNAs.

### 3.4. DDIT Inhibits Breast Cancer Cell Migration and Invasion

Hyaluronan has been shown to promote breast cancer cell invasion [22,23,25,26]. Thus, we investigated the effect of DDIT on breast cancer cells grown in collagen type I matrices. Like 4-MU, treatment of breast cancer cells with DDIT strongly inhibited their invasion (Figure 7A).

The highly invasive breast cancer cell line Hs578T expresses predominantly *HAS2* mRNA. DDIT treatment reduced cell migration by about 25% compared with untreated cells (Figure 7B). Of note, DDIT treatment of HAS2-depleted cells had no further effect on the motility, suggesting that DDIT targets HAS2-synthesized hyaluronan. Addition of exogenous hyaluronan of various molecular masses (1000 or 200 kDa) did not restore the migratory potential of DDIT-treated cells to the initial levels (Appendix A). The inhibitory effect of DDIT on motility was also verified in another triple-negative breast cancer cell line (MDA-MB-231; Appendix A).

Since breast cancer metastases often are located in the lung and are associated with a 60–70% patient mortality rate [48], we investigated the ability of DDIT to inhibit the migration of breast cancer cells exposed to lung microenvironment. Conditioned media from normal human lung fibroblasts (NHLF-2801-1) enhanced the migration of breast cancer cells, either in the absence or presence of 10% FBS, while addition of DDIT abolished the conditioned media-induced migration (Figure 7C). Moreover, in the presence of lung fibroblast-conditioned media, treatment with 4-MU or addition of Hermes-1, an antibody that specifically blocks hyaluronan/CD44 interactions, reduced the wound closure of the cell cultures to the same extent as DDIT treatment (Figure 7C). Thus, most likely, DDIT-mediated inhibition of the amount of hyaluronan and suppressed CD44-mediated Hs578T motility induced by lung-fibroblast conditioned medium.

### 3.5. DDIT Inhibits Breast Cancer Cell Proliferation

Inhibition of hyaluronan synthesis by DDIT inhibited the proliferation of Hs578T breast cancer cells after 6 and 9 days of culture (Figure 8A). As expected, the level of hyaluronan in 6 day-conditioned media of cultures treated with DDIT every third day, was reduced (Figure 8B). In accordance with this observation, the level of HAS2 protein, normalized to GAPDH, was reduced by about 40% at the same time point (Figure 8C).

Phase contrast images of the cells grown for up to 9 days in the absence or presence of the inhibitor did not reveal any alterations in cell morphology (Appendix A). Furthermore, no aging or apoptosis, as examined by β-galactosidase and FDA/PI staining, respectively, was observed after 6 days of treatment (Appendix A). Moreover, the level of phospho-S6, a marker of senescence, was unaltered and no cleavage of caspase-3, a marker of apoptosis, was observed (Appendix A). Thus, no evidence for DDIT-induced cell death was obtained.

To assess further the effect of DDIT on cell proliferation, we analyzed the cell cycle profiles of the cells using flow cytometry. Compared with untreated Hs578T cells, cells treated with DDIT for six days accumulated in the G0/G1 phase, and a corresponding decrease in the percentage of cells in the G2/M phase was noticed, indicating cell cycle arrest in G0/G1 (Figure 8D). Consistent with this observation, the level of the cell cycle regulatory proteins cyclin B1, was reduced by approximately 40% in DDIT-treated cultures, whereas the levels of cyclin E1 and cyclin D1, as well as the tumor suppressor p27, were unaffected (Figure 8E). It is thus possible that DDIT-mediated inhibition of hyaluronan suppressed cyclin B1 expression and thereby inhibited cell proliferation. The effect of DDIT on breast cancer cell proliferation was also verified using another triple-negative breast cancer cell line (MDA-MB-231; Appendix A)

### 3.6. DDIT Inhibits Mammosphere Formation of Breast Cancer Stem Cells

By interacting with its receptor CD44, hyaluronan regulates several aspects of cancer stem cell biology [27]. Thus, we next evaluated the effect of DDIT on breast cancer stem cell properties by culturing Hs578T cells in low-attachment conditions that enrich for the stemness phenotype. We observed that DDIT and 4-MU inhibited hyaluronan synthesis by about 40% and 90%, respectively, in breast cancer cell mammospheres (Figure 9A). This was paralleled by a reduction in breast cancer cell stemness, as indicated by reduced sphere forming efficiency (Figure 9B). Unexpectedly, the expression of the well-established stem cell markers *SOX2*, *OCT4* and *NANOG* were enhanced in mammospheres grown in the presence of DDIT or 4-MU (Figure 9C). The expression of hyaluronan synthases (*HAS1-3*) was induced after DDIT or 4-MU treatment, similar to the findings for cells grown in 2D, but this did not reach statistical significance (Figure 9D). Similar to the finding using 2D culture, the expression of *TMEM2* mRNA was reduced after hyaluronan synthesis inhibition by DDIT or 4-MU (Figure 9E). It is possible that in breast cancer spheres the expression of stem cell markers was compensatory increased after DDIT-induced (and 4-MU-induced) suppression of tumor cell self-renewal.

## 4. Discussion

A recent study of the Paramecium bursaria Chlorella virus CZ-2 HAS has provided insights into how HAS enzymes coordinate their substrates to catalyze glycosyl transfer and open a channel through the membrane to secrete hyaluronan, thereby coupling polymerization and translocation of the newly synthesized hyaluronan molecules [49]. However, the difficulties in solubilizing the mammalian multi-membrane embedded HASes in an active form for structural determination has impeded the development of small molecule inhibitors for mammalian HASes [41,50,51]. By screening candidate small molecule inhibitors of hyaluronan synthesis, such as substrate mimetics of HASes and natural products that competes with substrate binding or block polymer translocation, we identified DDIT, a novel, non-toxic and more potent inhibitor for hyaluronan synthesis than 4-MU.

The activities of HASes are regulated both at the transcriptional level and by post-translational modifications of HASes, as well as by dimerization [8,15,49]. DDIT and 4-MU showed similar suppressive capacities in several cell lines. Importantly, DDIT was found to be more than 10-fold potent than 4-MU (Figure 3E). DDIT was found to be active against both HAS2 and HAS3 enzymes which is consistent with the observation that cell lines that express both HAS2 and HAS3 (i.e., A549 and U-251 MG) produced less hyaluronan after treatment with DDIT. This conclusion is not surprising since HAS2 and HAS3 share about 70% amino acid sequence similarity.

The two-fold induction of *HAS1* and *HAS3* and the slight induction of *HAS2* in 2D and 3D cultures indicate that DDIT does not inhibit the production of hyaluronan by suppressing the transcription of mRNA for HAS isoforms. It is possible that the transcriptional induction of mRNA for *HASes* is a feedback mechanism for the cells to maintain a certain level of hyaluronan, which is important for cell proliferation [52]. 4-MU inhibits hyaluronan production by reducing the availability of UDP-GlcUA in the cytoplasm potentially by acting as a competitive substrate for UDP-glucuronosyltransferase leading to the formation of 4-MU glucuronide [34]. By analogy, it is possible that DDIT is converted to a metabolite, which in turn regulates hyaluronan synthesis. However, further studies are required to determine the precise mechanism by which DDIT acts in cells.

We observed that exogenously added hyaluronan could not rescue the lost migratory capacity of cells after DDIT or 4-MU treatment (Appendix A). This finding agrees with our previous studies [52], and suggests that the effect of endogenously synthesized hyaluronan cannot always be mimicked by exogenously added hyaluronan. Hyaluronan has important functions in the microenvironment and its size is a critical factor in defining its functional properties, that could be pro- or anti-tumorigenic. The hyaluronan in the extracellular matrix can be produced by tumor cells as well as the adjacent stromal cells [53,54]. Of note, tumor-associated hyaluronidase activity may degrade hyaluronan to shorter fragments promoting angiogenesis and tumor progression [55]. Triple-negative breast cancer displays high incidence of metastases and lung is a common metastatic area [56]. It should be noted that DDIT significantly reduced the expressions of *TMEM2*, encoding a main hyaluronidase, and CD44s at the mRNA level. However, the size of hyaluronan was not affected by DDIT treatment suggesting that DDIT does not inhibit migration and invasion through alterations in the molecular mass of hyaluronan (Figure 6B,C). Whether DDIT inhibits migration and invasion through hyaluronan-independent mechanisms is currently unknown and requires further studies. The importance of CD44-mediated cell adhesion for Hs578T cell migration is illustrated by our finding that migration induced by conditioned media of lung fibroblasts was inhibited, not only by DDIT, but also by the CD44 inhibitory antibody Hermes-1 (Figure 7C).

Since hyaluronan production and the cell cycle are interconnected (M. Mehić, et. al., Oncogenesis, 2017, DOI:10.1038/oncsis.2017.45 [17]), it is difficult to distinguish whether hyaluronan inhibition by DDIT causes cell cycle arrest or vice versa. The data of our study show that hyaluronan synthesis is inhibited by DDIT by as early as 12 h, whereas reductions at cell proliferation and cell cycle arrest are observed after 6 days of treatment. Therefore, it is likely that DDIT inhibits hyaluronan first and subsequently causes cell cycle arrest. Moreover, it must be noted that in a previous study from our group, knock-down of HAS2 in the same breast cancer cell line (Hs578T) reduced cyclin B levels while incubation of the cells with fragmented hyaluronan induced its expression (Yuejuan Li, et. al., Int J Cancer, 2007, DOI:10.1002/ijc.22550 [52]), suggesting that cyclin B expression lies downstream of hyaluronan signaling. The possibility that the observed cell cycle arrest is caused by hyaluronan-independent effects of DDIT remains an open question which will be examined in the future.

In 3D culture conditions, that select for cancer stem cell properties [57], DDIT suppressed hyaluronan synthesis and mammosphere formation (Figure 9). These results agree with the established role of hyaluronan in breast cancer cell stemness [27,58]. The unexpected upregulation of the stem cell markers *SOX2*, *OCT4* and *NANOG* after treating mammospheres with DDIT, may reflect a compensatory mechanism by which the cells in the spheres try to counteract the inhibitory effect of DDIT on cell proliferation. Similar stemness induction by an anti-cancer agent has been recently documented for 5-fluorouracil [59].

## 5. Conclusions

In conclusion, in this study we report the identification of a new small molecule inhibitor of hyaluronan synthesis, the thymidine analog DDIT, which is more potent than the widely used 4-MU, and significantly inhibits the aggressive phenotype of triple-negative breast cancer cells.

## Figures and Tables

**Figure 1 cancers-14-05800-f001:**
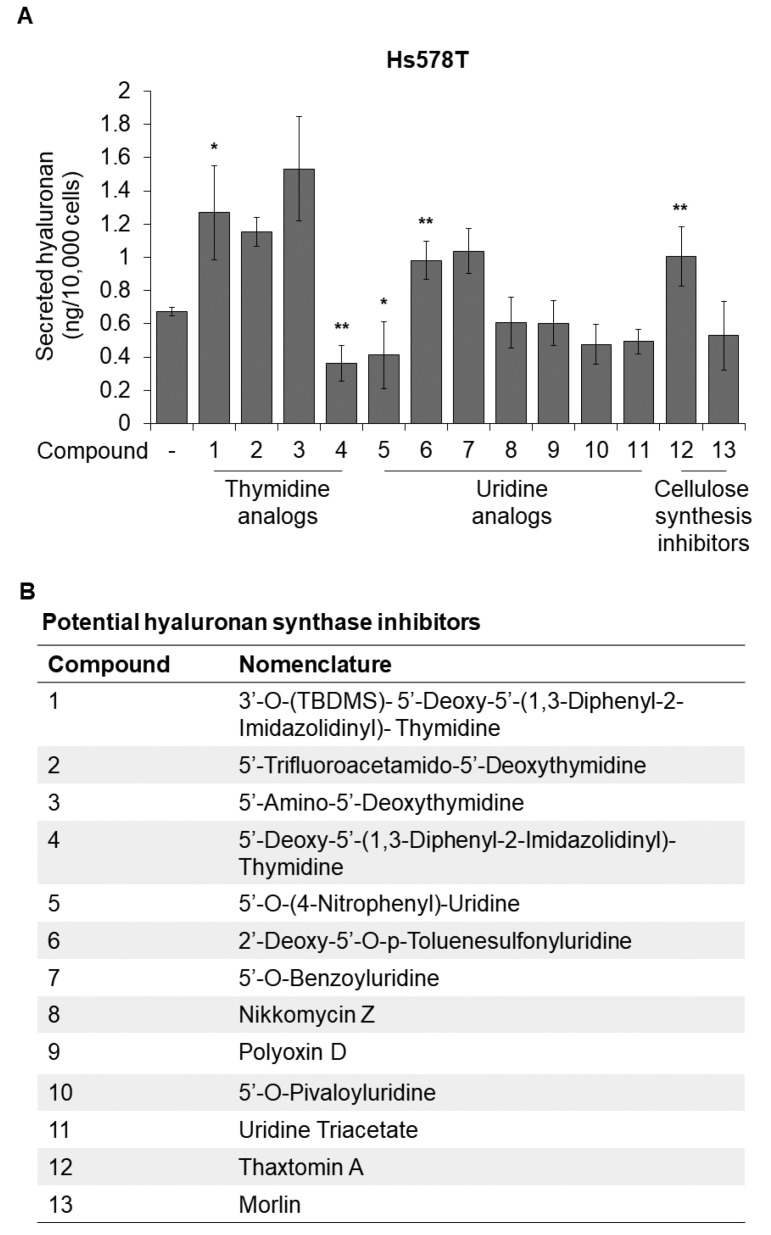
Screening for hyaluronan synthesis inhibitors. (**A**) Quantification of hyaluronan in the supernatant of Hs578T breast cancer cells after treatment with 20 μM of potential hyaluronan synthesis inhibitors for 24 h, in serum-free medium. DMSO was used as vehicle at a final concentration of 0.2%. (**B**) List of potential hyaluronan synthesis inhibitors. Statistical significance is depicted as asterisks (*, *p* < 0.05; **, *p* < 0.01).

**Figure 2 cancers-14-05800-f002:**
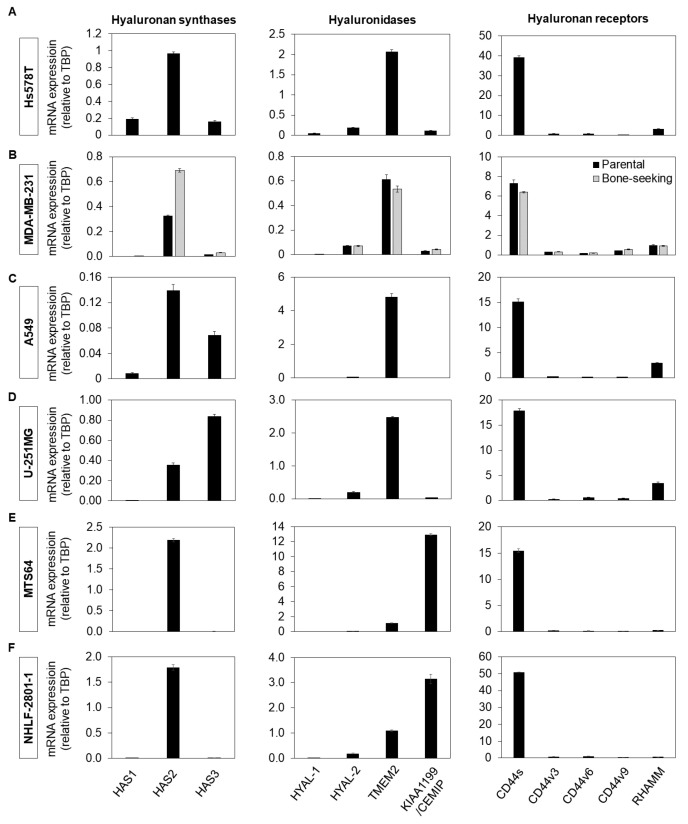
Expression of mRNA for hyaluronan synthases, hyaluronidases and hyaluronan receptors in various cell lines. mRNA expression analysis of hyaluronan synthases, hyaluronidases and hyaluronan receptors using RT-qPCR in Hs578T (**A**), MDA-MB-231 (parental and bone-seeking clone) breast cancer (**B**), A549 lung adenocarcinoma (**C**), U-251MG glioblastoma cells (**D**), and MTS64 dermal (**E**) and NHLF-2801-1 (**F**) lung fibroblasts.

**Figure 3 cancers-14-05800-f003:**
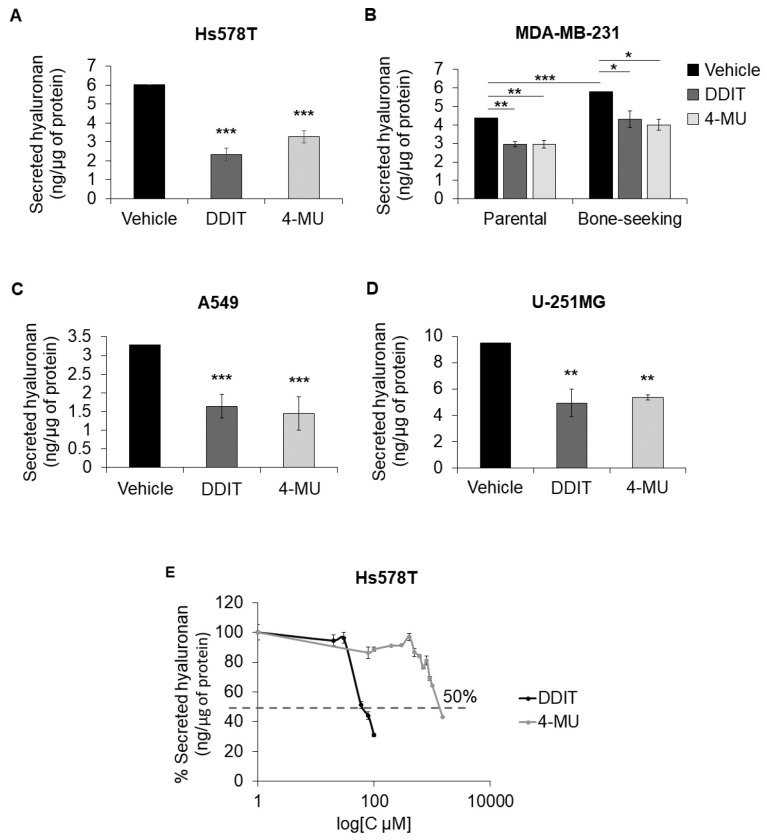
DDIT is a more potent inhibitor of hyaluronan synthesis than 4-MU. Quantification of hyaluronan secreted by Hs578T breast cancer cells (**A**), MDA-MB-231 parental and bone-seeking breast cancer cells (**B**), A549 lung adenocarcinoma cells (**C**) and glioma U-251MG cells (**D**) after treatment with vehicle, DDIT or 4-MU (1 mM) in serum-free medium. Hs578T, MDA-MB-231 and A549 were treated with 80 μM whereas U-251MG was treated with 100 μM DDIT. (**E**) Inhibition of hyaluronan synthesis after treatment with various concentrations of DDIT (0–100 µM) or 4-MU (0–1000 µM). Statistical significance is depicted as asterisks (*, *p* < 0.05; **, *p* < 0.01; ***, *p* < 0.001).

**Figure 4 cancers-14-05800-f004:**
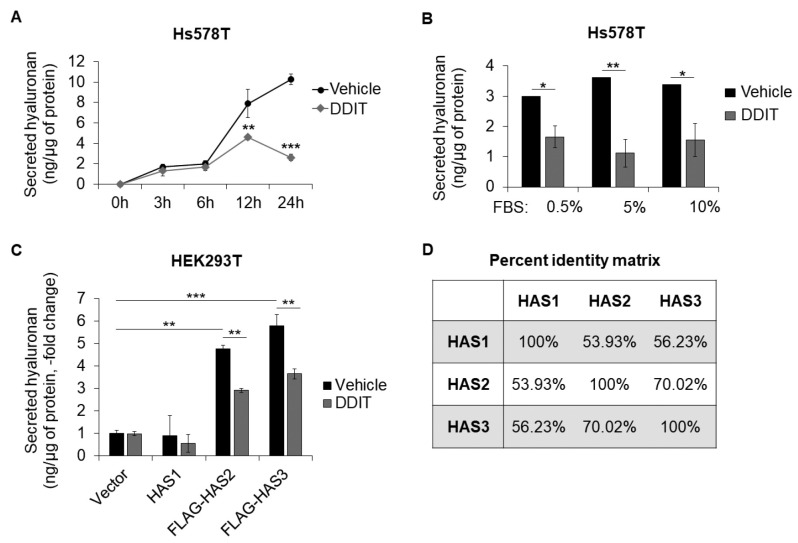
Effect of DDIT on hyaluronan synthesis by different HAS isoforms. (**A**) Hyaluronan secreted by Hs578T breast cancer cells after treatment with vehicle (0.04% DMSO) or DDIT (80 μM) in serum-free medium for 3, 6, 12 and 24 h. (**B**) Hyaluronan secreted after incubation with vehicle or DDIT in culture medium with serum (0.5, 5 or 10% FBS), for 24 h. (**C**) Hyaluronan secreted by HEK293T cells after overexpression of HAS1, FLAG-HAS2 or FLAG-HAS3 and incubation with vehicle or DDIT. (**D**) Comparison of amino acid sequence. Statistical significance is depicted as asterisks (*, *p* < 0.05; **, *p* < 0.01; ***, *p* < 0.001).

**Figure 5 cancers-14-05800-f005:**
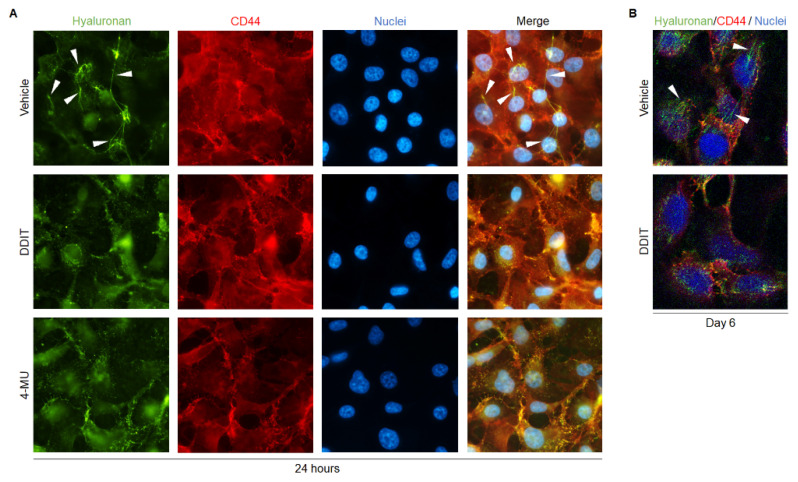
DDIT inhibits formation of hyaluronan cable-like structures. (**A**) Representative images of immunofluorescence staining for hyaluronan (green)/CD44 (red) in Hs578T cell cultures, 24 h after treatment with vehicle (0.04% DMSO), DDIT (80 μM) or 4-MU (1 mM) in serum-free medium. (**B**) Confocal imaging of hyaluronan (green)/CD44 (red)-stained Hs578T breast cancer cells after treatment with vehicle or DDIT in 10% FBS, for 6 days. Nuclei were stained with DAPI (blue). Arrow heads indicate hyaluronan cable-like structures. The images were captured with a 63× objective.

**Figure 6 cancers-14-05800-f006:**
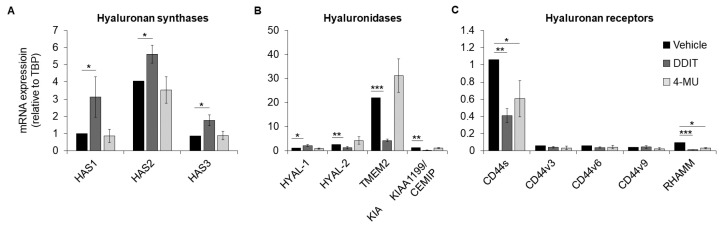
Effect of DDIT on the mRNA expression of hyaluronan synthases, hyaluronidases and hyaluronan receptors. The expression of mRNA for hyaluronan synthases (*HAS1*, *HAS2* and *HAS3*) (**A**), hyaluronidases (*HYAL-1*, *HYAL-2*, *TMEM2* and *KIAA1199/CEMIP*) (**B**) and hyaluronan receptors (*CD44s*, *CD44v3*, *CD44v6*, *CD44v9* and *RHAMM*) (**C**) was determined by RT-qPCR in Hs578T breast cancer cells after treatment with vehicle (0.04% DMSO), DDIT (80 μM) or 4-MU (1 mM) in serum-free medium, for 24 h. Statistical significance is depicted as asterisks (*, *p* < 0.05; **, *p* < 0.01; ***, *p* < 0.001).

**Figure 7 cancers-14-05800-f007:**
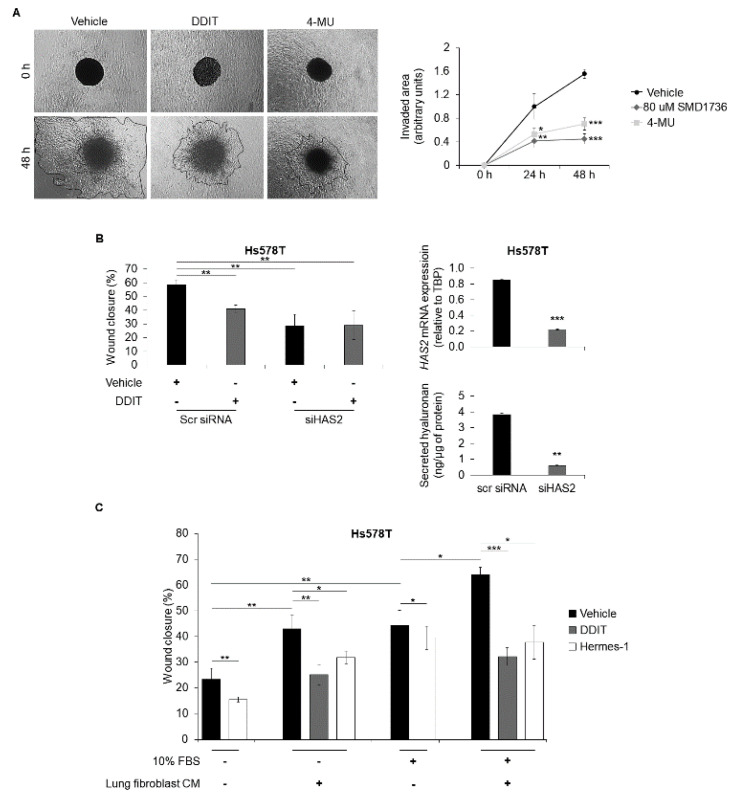
DDIT and HAS2 silencing inhibit breast cancer cell wound healing and DDIT inhibits migration of breast cancer cells in lung microenvironment. (**A**) Invasion of Hs578T breast cancer cells in collagen type I matrices after treatment with vehicle (0.04% DMSO), DDIT (80 μM) or 4-MU (1 mM) for 24 and 48 h, was determined by microscopy. Phase contrast images are shown from 0 and 48 h of the experiments. The images were captured with a 5× objective. (**B**) Wound healing of Hs578T breast cancer cell cultures after knock-down of *HAS2* with 40 nM siRNAs and treatment with vehicle (0.04% DMSO) or DDIT (80 μM) in medium containing 10% FBS, for 12 h. RT-qPCR for *HAS2* mRNA expression and quantification of secreted hyaluronan by Hs578T cells, after transfection with 40 nM siRNAs against *HAS2* or scrambled (scr) siRNA for 96 h are depicted. (**C**) Wound healing of Hs578T cell cultures after incubation with normal human lung fibroblast conditioned serum-free medium or medium containing 10% FBS and treatment with vehicle, DDIT or CD44 mAbs against Hermes-1 (blocks the binding of hyaluronan to CD44; 20 μg/mL) for 12 h. Statistical significance is depicted as asterisks (*, *p* < 0.05; **, *p* < 0.01; ***, *p* < 0.001).

**Figure 8 cancers-14-05800-f008:**
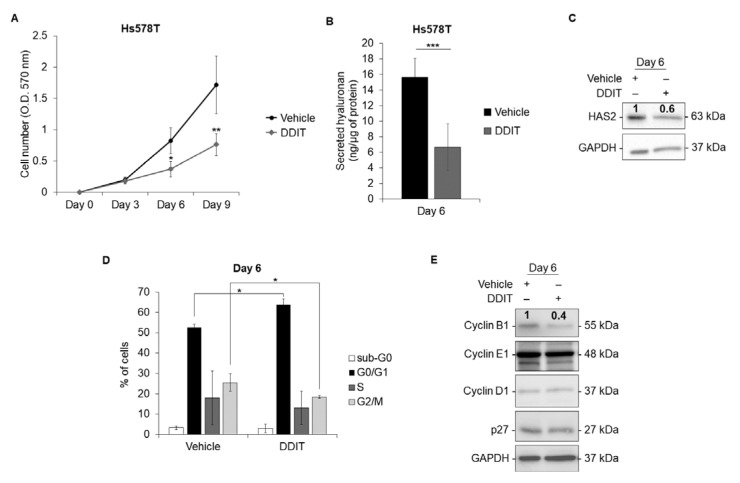
DDIT blocks cell cycle progression and inhibits breast cancer cell proliferation. (**A**,**B**,**D**) Proliferation of Hs578T breast cancer cells after treatment with vehicle (0.04% DMSO) or DDIT (80 μM) for 3, 6 and 9 days, in medium containing 10% FBS (**A**), hyaluronan secreted by Hs578T cells at day 6 (**B**), and analysis of Hs578T cell cycle progression with flow cytometry at day 6 (**D**). (**C**,**E**) Immunoblotting of HAS2 and GAPDH (**C**) and of cyclin B1, cyclin E1, cyclin D1, p27 and GAPDH (**E**) in total cell lysates of Hs578T, after treatment with vehicle or DDIT for 6 days, in medium containing 10% FBS. Statistical significance is depicted as asterisks (*, *p* < 0.05; **, *p* < 0.01; ***, *p* < 0.001). Original blots see Appendix A.

**Figure 9 cancers-14-05800-f009:**
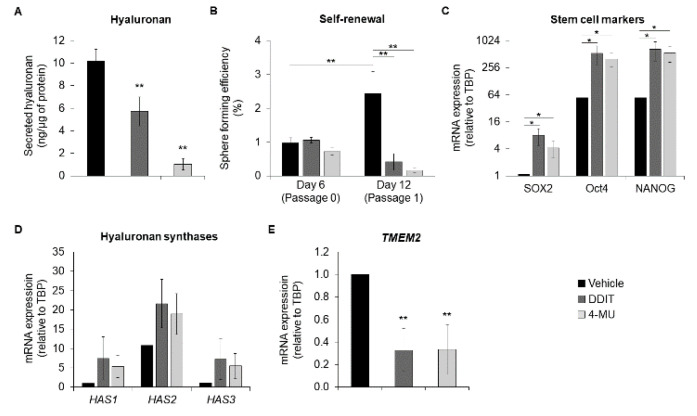
DDIT inhibits self-renewal of breast cancer stem cells. (**A**) Hyaluronan secreted by Hs578T mammospheres after 6 days of treatment with vehicle (0.04% DMSO), DDIT (80 μM) or 4-MU (1 mM). (**B**) Breast cancer stemness was assessed by quantification of Hs578T sphere forming efficiency after one passage (6 days per passage, totally 12 days), in the presence of vehicle, DDIT or 4-MU. (**C**) mRNA expression of stem cell factors (*SOX2*, *OCT4* and *NANOG*). (**D**) hyaluronan synthases (*HAS1*, *HAS2* and *HAS3*) and (**E**) *TMEM2*, were determined by RT-qPCR in Hs578T mammospheres grown for 6 days in the presence of vehicle, DDIT or 4-MU. Statistical significance is depicted as asterisks (*, *p* < 0.05; **, *p* < 0.01).

## Data Availability

The data presented in the study are available within the article and the Appendix A.

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
