# Peer review of "Identification of a Small Molecule Inhibitor of Hyaluronan Synthesis, DDIT, Targeting Breast Cancer Cells"

_cancers, 2022, doi:10.3390/cancers14235800_

Round 1

Reviewer 1 Report

This manuscript describes the identification of a new potential inhibitor of hyaluronan (HA) synthesis. The authors provide evidence that newly identified HA inhibitor suppresses breast cancer cell growth, migration, invasion, and self-renewal of cancer stem cells.

The effect of this inhibitor on cancer cell behavior has been well investigated in this study, and the findings may contribute to the development of anticancer drugs based on inhibition of HA synthesis. However, the authors need to pay more attention to the specificity and pharmacological effects of this inhibitor and address several issues as described below.

Major issues

1.      The thymidine analog DDIT may compete directly with donor substrates at the nucleotide binding site of HAS enzymes. To better clarify the inhibitory mechanism, the authors need to investigate the effect of this inhibitor on HAS enzyme activity using membrane fractions derived from HAS transfectants.

2.      To verify the specificity of this inhibitor, the authors need to examine its effect on the biosynthesis of other glycans, e.g., glycosaminoglycans such as heparan sulfates and chondroitin sulfates.

3.      DDIT treatment appears to induce cell cycle arrest at the G0/G1 phase by reducing cyclin B1 levels. In this regard, DDIT may exert broad pharmacological effects, possibly by affecting many biological events related to the cell cycle. Since HA synthesis and the cell cycle are closely related to each other, the authors need to discuss the possibility that DDIT may affect HA synthesis via cell cycle arrest.

4.      In page 13 and Fig. 9, the authors concluded that DDIT inhibits self-renewal of breast cancer stem cells. However, evaluation of sphere-forming capacity alone is not sufficient to assess self-renewal capacity of cancer stem cells. Therefore, the reviewer recommends that the authors revise the conclusion as follows: “DDIT inhibits mammosphere formation of breast cancer cells.”

Minor concerns

1.      Figure 6 which is currently placed on page 15 should be replaced below the description of the results on page 11.

Author Response

Reviewer #1

This manuscript describes the identification of a new potential inhibitor of hyaluronan (HA) synthesis. The authors provide evidence that newly identified HA inhibitor suppresses breast cancer cell growth, migration, invasion, and self-renewal of cancer stem cells.

The effect of this inhibitor on cancer cell behavior has been well investigated in this study, and the findings may contribute to the development of anticancer drugs based on the inhibition of HA synthesis. However, the authors need to pay more attention to the specificity and pharmacological effects of this inhibitor and address several issues as described below.

Major issues

  1. The thymidine analog DDIT may compete directly with donor substrates at the nucleotide-binding site of HAS enzymes. To better clarify the inhibitory mechanism, the authors need to investigate the effect of this inhibitor on HAS enzyme activity using membrane fractions derived from HAS transfectants.

Thank you for this important point. We performed non-radioactive in vitro hyaluronan synthase activity experiments as described in [DOI: 10.1074/jbc.M109.040386] with the modification that the end product was measured by means of hyaluronan ELISA-like assay as described in the material and methods section. However, the results were inconclusive since this assay was not sensitive enough. Unfortunately, we were unable to obtain the radioactive precursor (UDP 14C-GlcUA) needed for a more sensitive assay. Thus, we have not been able to address this point.

  1. To verify the specificity of this inhibitor, the authors need to examine its effect on the biosynthesis of other glycans, e.g., glycosaminoglycans such as heparan sulfates and chondroitin sulfates.

By performing 35S labeling of glycosaminoglycans in Hs578T cells, we observed that DDIT also inhibits the synthesis of sulfated glycosaminoglycans, a result suggesting that DDIT also acts on glycosaminoglycan synthesis mechanisms (Fig. S3C). Future studies will assess the role of DDIT in modulating high glycosaminoglycan concentrations in breast cancer.

  1. DDIT treatment appears to induce cell cycle arrest at the G0/G1 phase by reducing cyclin B1 levels. In this regard, DDIT may exert broad pharmacological effects, possibly by affecting many biological events related to the cell cycle. Since HA synthesis and the cell cycle are closely related to each other, the authors need to discuss the possibility that DDIT may affect HA synthesis via cell cycle arrest.

Since hyaluronan production and the cell cycle are interconnected (Mehić, et. al., Oncogenesis, 2017, DOI: 10.1038/oncsis.2017.45), it is difficult to distinguish whether hyaluronan inhibition by DDIT causes cell cycle arrest or vice versa. The data of our study show that hyaluronan synthesis is inhibited by DDIT by as early as 12 hours, whereas reductions in cell proliferation and cell cycle arrest are observed after 6 days of treatment. Therefore, it is likely that DDIT inhibits hyaluronan first and subsequently causes cell cycle arrest. The possibility that the observed cell cycle arrest is caused by hyaluronan-independent effects of DDIT remains an open question that will be examined in the future. This has now been discussed in a section in “Discussion”.

  1. On page 13 and Fig. 9, the authors concluded that DDIT inhibits the self-renewal of breast cancer stem cells. However, the evaluation of sphere-forming capacity alone is not sufficient to assess the self-renewal capacity of cancer stem cells. Therefore, the reviewer recommends that the authors revise the conclusion as follows: “DDIT inhibits mammosphere formation of breast cancer cells.”

Thank you for the comment, the appropriate changes have been introduced in the text.

Minor concerns

  1. Figure 6 which is currently placed on page 15 should be replaced below the description of the results on page 11.

Thank you for this comment the figure position has been corrected.

Reviewer 2 Report

The manuscript by Karalis et al., presents a novel inhibitor- DDIT- of hyaluronan synthesis. This inhibition leads to elimination of breast cancer cells aggressiveness by inhibiting proliferation, migration, invasion and cancer cells’ stemness. In this work the authors presented that, in several cancer cell lines, among hyaluronan synthases, hyaluronidases and hyaluronan receptors that HAS2, TMEM2 and CD44 are those that mainly expressed. Inhibitory effect of DDIT was observed in various serum conditions as well as in conditions with overexpression of HASes. In addition, DDIT affects hyaluronan structure and inhibits breast cancer cells’ migration and invasion in a HAS2- and CD44-dependent manner. DDIT inhibits also cancer cells’ proliferation via cyclin B1 inhibition and reduces TMEM levels in 2D and 3D cell cultures. Summarizing, the authors presented all the necessary evidence to support their hypothesis and the manuscript is well-written. The authors claimed that hyaluronan synthesis inhibition led to reduction of cell proliferation via cyclin B1. A point that could be figured out is whether silencing of hyaluronan synthesis downregulates cyclin B1 levels or if DDIT inhibits cyclin B1 via a hyaluronan-independent mechanism. A minor comment in Fig 2. Is that the KIAA1199/CEMIP labeling is missing.

Author Response

Reviewer #2

The manuscript by Karalis et al. presents a novel inhibitor- DDIT- of hyaluronan synthesis. This inhibition leads to elimination of breast cancer cells aggressiveness by inhibiting proliferation, migration, invasion and cancer cells’ stemness. In this work the authors presented that, in several cancer cell lines, among hyaluronan synthases, hyaluronidases and hyaluronan receptors that HAS2, TMEM2 and CD44 are those that mainly expressed. Inhibitory effect of DDIT was observed in various serum conditions as well as in conditions with overexpression of HASes. In addition, DDIT affects hyaluronan structure and inhibits breast cancer cells’ migration and invasion in a HAS2- and CD44-dependent manner. DDIT inhibits also cancer cells’ proliferation via cyclin B1 inhibition and reduces TMEM levels in 2D and 3D cell cultures. Summarizing, the authors presented all the necessary evidence to support their hypothesis and the manuscript is well-written. The authors claimed that hyaluronan synthesis inhibition led to reduction of cell proliferation via cyclin B1. A point that could be figured out is whether silencing of hyaluronan synthesis downregulates cyclin B1 levels or if DDIT inhibits cyclin B1 via a hyaluronan-independent mechanism. A minor comment in Fig 2. Is that the KIAA1199/CEMIP labeling is missing.

Thank you for the valuable comments. Silencing of HAS2 in this cell line has been shown to reduce the expression of cyclin B1 in a previous study from our group, which is in agreement with the findings of the current study.

We agree with the comment that DDIT can inhibit cyclin B expression via hyaluronan-independent mechanisms but providing experiments to demonstrate this point is tedious and will be investigated further in the future. Nevertheless, it must be noted that in a previous study from our group, the knock-down of HAS2 in the same breast cancer cell line (Hs578T) reduced cyclin B levels while incubation of the cells with fragmented hyaluronan induces its expression (Li, et. al., Int J Cancer, 2007, DOI: 10.1002/ijc.22550), suggesting that cyclin B expression lies downstream of hyaluronan signaling.

The point regarding Fig. 2 has been corrected.

Reviewer 3 Report

The manuscript by Karalis et. al; summarizes the results obtained from the identification of a small molecular inhibitor of hyaluronan synthesis, DTT targeting the breast cancer cells. A  number of studies have demonstrated that hyaluronan regulates cell migration and invasion as well as tumor growth and progression (in vitro and in vivo). A well characterized small molecule inhibitor of hyaluronan is 4-MU. It has been reported that 4-MU has modest potency towards the breast cancer cells which limits its therapeutic utility. Here, the authors have reported the identification of DDIT as a potential inhibitor of hyaluronan synthesis and have provided a characterization of its functional properties. To assess the effects of DDIT on hyaluronan synthesis, the authors have selected various cell lines which synthesize hyaluronan. Using RT-PCR, they determined their mRNA ex pression of molecules involved in signaling. The authors found that DDIT is more potent inhibitor of hyaluronan synthesis than 4-MU. They also found that DDIT inhibits breast cancer migration, invasion  and proliferation. Overall, a very well drafted manuscript and all the experiments have been performed to support the hypothesis.

There are minor typos which can be corrected:

1)      Line 123, please remove ref and add # for the Qiagen RNeasy mini kit to have the consistency throughout with the catalog numbers.

2)      Line 187, please change two hundred fifty thousand to 250,000 or 250K since the cell numbers are usually mentioned in numeric value.

3)      Line 281, please write cell number as 800,000 or 800k to keep it uniform through the manuscript in numerical value

Author Response

Reviewer #3

The manuscript by Karalis et. al; summarizes the results obtained from the identification of a small molecular inhibitor of hyaluronan synthesis, DTT targeting the breast cancer cells. A number of studies have demonstrated that hyaluronan regulates cell migration and invasion as well as tumor growth and progression (in vitro and in vivo). A well characterized small molecule inhibitor of hyaluronan is 4-MU. It has been reported that 4-MU has modest potency towards the breast cancer cells which limits its therapeutic utility. Here, the authors have reported the identification of DDIT as a potential inhibitor of hyaluronan synthesis and have provided a characterization of its functional properties. To assess the effects of DDIT on hyaluronan synthesis, the authors have selected various cell lines which synthesize hyaluronan. Using RT-PCR, they determined their mRNA expression of molecules involved in signaling. The authors found that DDIT is more potent inhibitor of hyaluronan synthesis than 4-MU. They also found that DDIT inhibits breast cancer migration, invasion and proliferation. Overall, a very well drafted manuscript and all the experiments have been performed to support the hypothesis.

There are minor typos which can be corrected:

  1. Line 123, please remove ref and add # for the Qiagen RNeasy mini kit to have the consistency throughout with the catalog numbers.

Thank you for this comment. The appropriate changes have been introduced.

  1. Line 187, please change two hundred fifty thousand to 250,000 or 250K since the cell numbers are usually mentioned in numeric value.

Thank you for this comment. We have revised the text.

  1. Line 281, please write cell number as 800,000 or 800k to keep it uniform through the manuscript in numerical value.

Thank you for this comment. We introduced the changes in the manuscript.

Round 2

Reviewer 1 Report

The authors fully responded to all my concerns raised and made the necessary changes to the manuscript.